# Physiological Electric Field: A Potential Construction Regulator of Human Brain Organoids

**DOI:** 10.3390/ijms23073877

**Published:** 2022-03-31

**Authors:** Xiyao Yu, Xiaoting Meng, Zhe Pei, Guoqiang Wang, Rongrong Liu, Mingran Qi, Jiaying Zhou, Fang Wang

**Affiliations:** 1Department of Histology & Embryology, College of Basic Medical Sciences, Jilin University, Changchun 130021, China; yxy19@mails.jlu.edu.cn (X.Y.); mengxt@jlu.edu.cn (X.M.); liurr20@mails.jlu.edu.cn (R.L.); jiaying21@mails.jlu.edu.cn (J.Z.); 2Department of Neurology, School of Medicine, The City College of New York, 160 Convent Avenue, New York, NY 10031, USA; topeizhe@hotmail.com; 3Department of Pathogenic Biology, College of Basic Medical Sciences, Jilin University, Changchun 130021, China; wanggq20@jlu.edu.cn (G.W.); qimr@jlu.edu.cn (M.Q.)

**Keywords:** brain organoids, physiological electric field, neurogenesis, neuronal differentiation, neural network

## Abstract

Brain organoids can reproduce the regional three-dimensional (3D) tissue structure of human brains, following the in vivo developmental trajectory at the cellular level; therefore, they are considered to present one of the best brain simulation model systems. By briefly summarizing the latest research concerning brain organoid construction methods, the basic principles, and challenges, this review intends to identify the potential role of the physiological electric field (EF) in the construction of brain organoids because of its important regulatory function in neurogenesis. EFs could initiate neural tissue formation, inducing the neuronal differentiation of NSCs, both of which capabilities make it an important element of the in vitro construction of brain organoids. More importantly, by adjusting the stimulation protocol and special/temporal distributions of EFs, neural organoids might be created following a predesigned 3D framework, particularly a specific neural network, because this promotes the orderly growth of neural processes, coordinate neuronal migration and maturation, and stimulate synapse and myelin sheath formation. Thus, the application of EF for constructing brain organoids in a3D matrix could be a promising future direction in neural tissue engineering.

## 1. Introduction

Brain organoids are created by three-dimensional (3D) stem cell cultures, forming a miniature organ resembling the brain. These in vitro culture systems are derived from culturing a population of self-renewing, pluripotent stem cells (PSCs) and inducing them to differentiate into neural-specific cell types, mimicking the spatial organization of the brain and reproducing some brain functions. Brain organoids recreate the components of the human brain, not only at the cellular level but also in terms of 3D tissue structure and in a developmental trajectory [1]. As an important 3D organ model system, brain organoids are potentially useful in the following ways: (1) representing early brain development, they provide an in vitro platform, facilitating the in-depth study of neurodevelopment; (2) when culturing patient-derived cells (PDCs) using this platform, this could be considered a highly physiologically relevant system for studying the pathological mechanisms of mental illnesses and other neurological disorders [1,2,3]; (3) human cerebral organoids could also be used as a therapeutic drug screening model when seeking solutions for neurological diseases, such as Alzheimer’s disease and Creutzfeldt-Jakob disease [4,5]. In particular, the neurons grown in these brain organoids possess some of the in vivo functional characteristics of human neurons, such as the resting membrane potential, some firing patterns, and the plasticity of specific subtypes of neurons [6]. Additionally, the cerebral organoid has also been used broadly used to recreate particular brain structures, such as the midbrain [7,8], hippocampus [9], hypothalamus [10], and cerebellum [11]. These results demonstrate the advantages of brain organoid technology for research simulating the nervous system both physically and organizationally.

Thus, creating brain organoids in artificial environments has aroused the interest of many researchers. To obtain structures that simulate a particular brain region, the most straightforward approach could be generating a similar microenvironment and mimicking the procedure of neural differentiation in a dish. In 2013, Lancaster et al. [12] developed a culture system made up of multiple chemical inducers to produce a human PSC-derived 3D cerebral organoid. Up until now, the generally accepted and most effective construction method has been to introduce human PSCs (hPSCs) into embryoids first and then transfer them to an extracellular support matrix (such as Matrigel) to facilitate cell growth in a 3D environment [13]. At different neural developmental time points, an induction medium containing different chemical cues that support neural development, including growth factors, hormones, and neurotrophins, is added to the culture system [12,14]. Alternatively, using small molecular or genetic modifications to regulate well-known neurodevelopmental signaling pathways can also somehow induce neural differentiation and the formation of brain organoids [15,16,17].

Besides chemical inducers, in vivo neural development also relies on physical cues in extracellular environments, such as mechanical (e.g., rigidity and strain), electrical, and topographical cues, to induce and regulate development. These physical cues could synergistically interact with chemical cues. It has been reported that the migration, differentiation, neurite growth, and myelination of neural cells are all regulated by the stimulation of physiological electrical signals [18,19]. In 2019, Bertucci, C.’s study further confirmed that the intensity, location, and duration of physiological electric fields (EFs) could influence cell behaviors during neural development [20]. This suggests that electrical cues could be a potential construction regulator of human brain organoids.

This review will explore the role of bioelectrical stimulations in regulating neural development, neural lineage cell behaviors, and the formation of a neuronal network. Furthermore, we discuss the potential of applying physiological electrical stimulations during the process of culturing 3D brain organoids in a dish, highlighting all novel construction strategies and important advances in this unique field of tissue engineering.

## 2. Traditional Approaches to Obtain Brain Organoid

### 2.1. Following the In Vivo Procedures of Neural Development as a Strategy for Constructing Brain Organoids

Learning from in vivo neurogenesis mechanisms or even mimicking key features of early brain development could be a productive strategy for constructing brain organoids. For instance, both brain development and organoids could be initiated from hPSCs with excellent self-organization ability [21,22]. Brain organoids could represent 30–100% of the variable efficiency of polarized neural ectoderm formation [18]. Human forebrain organoids could also be induced to express neural developmental guidance cues such as sonic hedgehog (SHH) in gradient patterns that are also secreted by floor-plate cells during neural plate development [23].

Another reason for mimicking the procedure of neural development to construct brain organoids is to meet the urgent need for studying human neurodevelopment and diseases. As an in vitro study model, brain organoids may need to accurately represent the early neural developmental trajectories: neural induction, the proliferation of neural stem cells (NSCs), cell migration, neuronal differentiation, axon guidance, intercellular connections and adhesion, inter-neuronal connections, synapse formation, myelin sheath formation, and the functional development of established neural networks [24]. Based on the need for each area of study, these processes may need to be isolated or overlapped.

In almost all brain organoids, neural lineage cells, including NSCs, radio glial cells, intermediate precursor cells, neurons, astrocytes, and oligodendrocytes emerge in a time sequence that resembles embryonic brain development [25]. For example, cortical development follows the “inside-out” pattern: neurons in layers V and VI, which are located in the deeper cortical layers, will mature first; neurons that are located in the upper layers (II–IV) mature after the deeper layers; most of the glial cells (astrocytes and oligodendrocytes) are the last to mature. During the culture of cortical organoids, neuronal markers expressed in the deeper cortical layers form at an earlier stage, while neurons expressing the upper layers’ markers are generated later [12].

In most cell culture conditions, the rate of neuronal differentiation for both ESCs/PSCs and NSCs is much lower compared with in vivo cortical development at similar time phases [26,27,28]. When constructing brain organoids that simulate in vivo the neural developmental process, we need to adjust the timeline of culture that corresponds to express regulating factors, then we need to initiate neuronal differentiation correctly and establish an orderly induction scheme for glial cell differentiation. Therefore, in the process of constructing brain organoids, different cell types are generated in the correct order and migrate to their proper situation, then form themselves into a well-organized 3D architecture.

Until now, following this construction principle, brain organoids constructed in the laboratory could be categorized into two groups: (1) brain organoids that have been pre-patterned, which could be cultured by using multiple patterning factors to guide cells that are differentiated into specific neural tissue formations [29,30,31]; (2) un-patterned brain organoids that contain spontaneously differentiated neuronal cells. Their formation relies on ESC/PSC’s own morphogenesis; therefore, they encompass a variety of morphological structures with a mixture of features seen in developing brain subareas, such as the forebrain, midbrain, hindbrain, retina, and choroid plexus [8,12,32,33,34].

### 2.2. Challenges for Constructing Brain Organoids

Brain organoids with or without certain structural patterns are still not able to completely capture the regional complexity, cellular diversity, and circuit functionality of any brain region. This presents researchers with three challenges. First, it could be difficult to form a multi-layered brain cytoarchitecture with diverse cell types and complex neuronal connections. Brain organoids that are cultured from human stem cells can only form a simple hierarchical structure in their early stage, but then fail to differentiate into a more complex multi-layered tissue architecture that resembles the embryonic human brain. The size of cultured brain organoids is also tiny: their dimensions are generally only around 4–6 mm [12,35,36]. Their architecture is quite different from the normal brain, which limits their applications to a whole range of developmental disease models or for acting as a replacement for a rodent model for neuronal drug discovery. Second, the formation of microvascular networks has become one of the major bottlenecks restricting the development of brain organoid construction techniques. Potential methods to create a functional vascular-like system with hETV2-induced endothelium cells have been reported by Bilal Cakir et al. [34]. Learning from relevant fields, other strategies for in vitro vascularization include printing endothelial cells as a template using bioprinting technology [37,38], or inducing vesicular fibroblast self-organization in vitro with the use of vascular endothelial growth factor (VEGF) [39]. The potential vascularization of brain organoids brings hope that it will be possible to create larger neural tissue chunks with a homogeneous nutritional supply and produce a better drug-testing system with an increased penetration rate. Third, some specific cell types are difficult to produce in brain organoids when following the general culture protocol. For instance, microglia cells, which also regulate neuronal differentiation, synapse formation and elimination [40,41,42], are rarely produced in brain organoids. In 2018, Paul Ormel et al., reported a method that could stimulate the maturation of microglia in a 3D context, suggesting that the mesodermal progenitors in brain organoids are a useful source of microglia-like cells within cerebral organoids [43]. Fourth and most importantly, although neurons in brain organoids can somehow form synaptic connections with each other, creating complete, well-organized, and functioning neural circuits with mature synapses is still a great challenge [44]. Since the neurons in organoids lack specific hierarchical dendritic structures and axon projection patterns, it is difficult to form myelin sheaths and establish well-developed neural networks [45]. Recently, D. Kacy Cullen et al., created tissue-engineered transplanTable 3D axon tracts [46]. Additionally, by inducing the genesis of myelinating oligodendrocytes, these “oligocortical spheroids” provide myelin compaction by week 30 of culture [47]. Generally, an ordered tissue structure and a well-organized neural network in brain organoids are essential to achieve these physical designs.

## 3. Could Adding EFs Enhance the Efficacy of Constructing Human Brain Organoids?

### 3.1. Using EFs Influences the Formation of Brain Organoids: Thinking Differently

In the microenvironment, both chemical cues and physical signals are known to influence cell behavior, thus determining neural development. The most direct evidence for this is that bioelectrical signals play a decisive role in the development of certain organs during embryonic development. When applying a specific voltage gradient from the back to the tail of amphibian tadpoles, extra “eye-spots” were induced to develop outside the head area. This was the first time that scientists artificially initiated the creation of a new organ at a certain location by altering an organism’s endogenous voltage difference [48].

During embryonic neural development, PSCs within the nervous system follow a spatial-temporal organized trajectory that must be generated, migrating to their terminated locations, maturing, and forming functional networks. Within the same process, these cells are self-organized to form a 3D structure with unique structure patterns. This process includes the following developmental events: neural induction, cell proliferation, migration, differentiation, axon guidance, myelin formation, synaptic formation, and neural network formation [18]. It has been reported that the migration, differentiation, protuberance growth, and myelination of nerve cells are not only regulated by chemical signals and intercellular contact but are also closely related to EFs during the development of nerve tissue (Figure 1) [18,49,50,51,52,53,54]. This suggests that biophysical regulators, particularly EFs, could be used to manipulate the process of neural development and to adjust the tissue architecture in a well-controlled culture system; therefore, they play a critical role in the in vitro construction of brain organoids.

To discuss the potential applications of EFs in constructing brain organoids, we need to begin with the role of endogenous EFs during neural development.

### 3.2. Endogenous EFs Regulate Neural Development In Vivo

Spatial and temporal cues influence neuronal tissue developments in both physiological and pathophysiological processes. Processes, such as neural tube formation and neural injuries, could generate ion flow and induce regional endogenous EFs [55].

Endogenous EFs occur in the cytoplasm and extracellular space. The strength of this form of EF can vary from a few mV/mm to hundreds of mV/mm [26]. During embryonic development, they appear in the inner side of the nerve plate and the nerve plica. This phenomenon also occurs in the lateral walls of neural tubes and in the tails of developing amphibian and bird embryos [27]. In the process of neural tube formation, blocking endogenous physiological EFs can cause neural development defects [56].

Endogenous EFs not only exist during early embryonic development but are also a regulator in the developing and adult cortex. Cao et al., measured the endogenous electric current that flows in the walls of the lateral ventricle along the rostral migratory pathway in adult mice and identified that neuroblasts migrate along this pathway [57]. Ohtaka-Maruyama et al., described electrical activity influencing neuronal migration during cortical development [58,59]. Based on these discoveries, Vera P. Medvedeva and Alessandra Pierani proposed that the developing cortex is a stratified structure that exhibits variable electric strength. During cortical development, the formation of the layered structure relies on the highly organized radial migration of newly born cells, which include glutamatergic neurons. Medvedeva et al., indicated that the subplate (SP) and the marginal zone (MZ) are potentially highly charged compared to the relatively low EF in the intermediate zone (IZ) and the cortical plate (CP). As a result, it generates ion flow and voltage differences among the different developing zones. Thus, endogenous electric currents work together with chemical cues, regulating the migration of glutamatergic neurons, orienting neuron polarization, and eventually driving the corresponding morphological changes: axon initiation under the SP and dendritogenesis in the MZ [60].

From these observations, we can infer that the generation of physiological EF may be closely related to the geometric changes in tissue morphology during development. Following that, the variations of EF accurately induce the direction of cell migration and lead to morphological changes in neural lineage cells, regulating the process of neural development and neurogenesis, which also contributes to spatial patterning maturation in the adult brain (Table 1).

**Table 1 ijms-23-03877-t001:** Endogenously generated bioelectric currents play a key role in important biological processes of brain formation.

Main Results	Species	References
Xenopus embryos maintain an inwardly positive electrical potential across their skin throughout most of their early development.	xenopus	[58]
EF exists in the inner side of the nerve plate and nerve folds and the side-wall of the neural tube during embryonic development. In the process of neural tube formation, blocking endogenous EF will cause neural development defects.	axolotl	[59]
Altering the internal field results in defects in the tail, limb bud, and head development.	chick	[27]
The physiological EFs direct and stimulate the migration of the SVZ neuroblasts along the rostral migratory path.	mouse	[61]
Subplate neurons extend neurites toward the ventricular side of the subplate and form transient glutamatergic synapses.	mouse	[55]
An electrically active boundary organizes neuronal migration during cortical development.	mouse	[60]
Cells manifest high electrical activity as they establish afferent and efferent synaptic connections within the developing cortex.	mouse/human	[62]
Ion flow and voltage difference are generated among different zones in the brain cortex.		

## 4. EFs Affect the In Vitro Behavior of Neural Cells

### 4.1. Types of Exogenous EFs

Evaluating cell behavior under in vitro physiological EFs is a rapidly growing field of study due to the significant advantage of creating and controlling both molecular and physical environments. Exogenous EFs are generated by an external power source and are usually applied to biological cells/tissues via electrodes. In these tests, extracellular electrical stimulations include the following types of elements: (1) direct current (DC), (2) alternating current (AC), (3) pulsed current (PC), (4) two-phase current (BEC), and (5) electromagnetic fields (EMFs) [18].

Directional ion flow generates endogenous EFs that are basically DC voltage gradients; thus, DC EF is the most common mode when applying EFs in vitro [57]. However, the application of a constant current in vitro may lead to the accumulation of cytotoxic by-products at both ends of the electrodes. Although the application of salt bridges reduced the entry of these by-products into the culture medium, using salt bridges correspondingly increases the electrical resistance, generating excess heat [63], which is also harmful to the culture. AC or PC EF application patterns are defined by their frequency (in Hz) and pulse width (in ms). They can balance the charge in the cathode and anode periodically by reversing the polarity of the electrodes; consequently, this reduces the cytotoxic by-products [20]. However, this specific application pattern may need a process-control system to manipulate. Therefore, the design of in vitro physiological EF application facilities still needs to be optimized.

Generally, electric fields and magnetic fields are linked together and unified. They can both be transformed into each other through “movement and change”. Changing EFs with moving currents and charges produce magnetic fields, while moving magnetic fields produce EFs as well. In both cases, the EMFs here are included in one of the types of EFs and are also known to regulate the proliferation, differentiation, and neurite outgrowth of NSCs [64,65,66]. For example, EMF plays a positive role in adult neurogenesis [67,68], and exposure to EMF (50 Hz, 0.4 mT) significantly enhanced the proliferation of embryonic brain-derived hippocampal NPCs [69].

### 4.2. EFs Induce the Directional Migration of Neural Lineage Cells In Vitro

EF plays an important role in regulating the migration direction and speed of neural lineage cells such as NSCs/NPCs, neurons, and oligodendrocytes. In vitro experiments demonstrated that both adult- and embryo-derived mice NSCs/NPCs or ESC-derived NSCs showed obvious electrotaxis under different intensities of EF, with directed cell migration from the negative pole toward the positive pole (Figure 2) [63,70,71]. Low-field-intensity DC EFs efficiently direct NSC movement (toward the cathode). When the electric field intensity increases from 50 mV/mm to 100 mV/mm, the migration distance increases significantly. After increasing the EF intensity up to 250~437 mV/mm, the electrical stimulation will not only initiate a directional migration but can also significantly promote the orientation and displacement of both embryonic and adult NPCs [70]. DC EFs can also guide the migration of isolated newborn hippocampal neurons from hypocotyl explants to the cathode. The direction of neuron migration can be reversed by reversing the EF vector [72,73].

Recent studies showed that oligodendrocyte progenitor cells (OPCs) have a similar ability: when exposed to an EF of physiological intensity, OPCs present significant electrotaxis. Zhu et al., demonstrated that OPCs from the brains of neonatal Sprague-Dawley rats showed marked electrotaxis toward the negative pole. Their directional migration depended on the key subunit of integrin receptors: β1 integrin. The directedness and displacement of cathodal migration increased significantly when the EF strength increased from 50 to 200 mV/mm [74].

The influence of EF on the migration direction, migration rate, and orientation are variable and are closely related to EF strength, species, cell types, or microenvironments [63,71,72]. For example, although most of the neural lineage cells prefer to migrate from anode to cathode, hiPSCs [75] and chicken Schwann cells migrated to the anode in EFs [76].

### 4.3. EFs Regulate In Vitro Neuronal Differentiation

Besides cell migration, cell proliferation and differentiation are also regulated by EFs [77]. It has been well documented that EFs either promote or inhibit cell proliferation, depending on the cell type and exposure conditions [78,79,80,81]. Here, we mainly discuss EF-regulated neuronal differentiation. This important event in neural development has been identified in different types of stem cells but has only been reported in recent years. Masahisa Yamada et al., were the first to report that electrical stimulations are critical to the fate choice of mice ESCs. In this report, EFs (125 mV/mm–500 mV/mm) were applied to stimulate the differentiation of ESC-formed embryoid bodies in vitro, and these electrical stimulations significantly promoted the cell differentiation toward neurons during the embryonic body stage [82]. Subsequently, electrical stimulus-induced neuronal differentiation has been demonstrated in multiple cell types. Ariza et al., demonstrated that applying an EF strength of 437mV/mm can promote the differentiation of adult hippocampal NPCs into neurons [83]. Park et al., performed 250mV/mm electrical stimulation on human mesenchymal stem cells (hMSCs) and identified that electrical stimulation can significantly induce the differentiation of hMSCs into nerve cells only by the synergistic action of the overexpressed exogenous Nurr1 gene [75]. Short periods (2 days) of electrical stimulation (at 0.53 or 1.83 V/m for 10 min/day) enhanced neurite growth and the expression levels of neuronal markers (such as βIII tubulin and neuronal nucleus (NeuN)), and increased intracellular Ca^2+^ during stimulation [73]. The proliferation rates and axon growth of PC12 cells could be enhanced by electrical stimulation; enhancing the intensity of the electrical stimulation increases the cell number significantly [53]. The efficiency of EF-induced neuronal differentiation was dependent upon stimulation parameters. It is noteworthy that the diversity in the fate choice of NSCs that is induced by electrical stimulation may also be related to differences in the microenvironment. Electrical stimulation cooperated with different synergistic conditions, such as growth factors, the extracellular matrix, etc., resulting in diverse differentiated cell types and ratios (Table 2). Many studies have shown that physiological EF can affect cell proliferation, differentiation, and migration by activating voltage-sensitive genes, thereby modifying the expression of a series of genes in their downstream signal transduction pathways. Because voltage-gated ion channels could also be used to trigger secondary messages and modify intracellular signaling cascades, EFs might be combined with other gene expression regulators, such as growth factors or the extracellular matrix in the microenvironment, thus producing synergies.

**Table 2 ijms-23-03877-t002:** The efficiency of EF-induced neuronal differentiation was dependent upon the stimulation parameters.

Cell Type	Tuj1%	MAP2%	GFAP%	Oligo%	Intensity	Time	Species	References
NPCs	16.9 ± 5.3	—	61.6 ± 2.7	10.1 ± 0.7	115 V/m	2 h/day	mouse	[70]
NSCs	~2.3	~1.5	—	—	100 Hz	7 days	mouse	[84]
NPCs	42	—	15	—	437 mV/mm	16–24 h/day	rat	[79]
MSCs	↑ 4.5 folds	↑ 4 folds	—	—	250 mV	1000 s	human	[80]
NSCs	↑~2.5 folds	—	↑~1.5 folds	—	20–30 μA	—	mouse	[81]
NPCs	18.3 ± 12.0 — —	— — —	— 54.9 ± 23.5 —	— — 0.7 ± 0.2	10 Hz 1 Hz 10 Hz	3 DIV 14 DIV 14 DIV	porcine	[82]
iPSCs	—	29	—	—	30 μA	10 min	human	[74]
NPCs	20.9	—	69.4	29	300 mV/mm	48 h	Mouse (E13.5)	[75]
NSCs	79.5	—	11.6	—	75 mV	7 days	human	[53]
NSCs	the ratio of neurons to astrocytes 1:2				0.1~10 Hz	1/7/14/21 days	mouse	[85]

### 4.4. EFs Promote Axon Outgrowth and Alignment

The outgrowth of neurites is a fundamental procedure during neural development and regeneration that is related to nerve fiber projection, branching, synapse formation, and neuron maturation [18]. The guiding effect of electrical stimulation on neurite growth and alignment has been well documented (Table 3).

**Table 3 ijms-23-03877-t003:** The guiding effect of electrical stimulation on neurite growth and alignment.

Species	Cell Type	EF Type	Intensity	Time	Main Results	Reference
Chicken	embryo dorsal root ganglion	DC	70~140 mV/mm	20 h	The growth rate of the protrusion on the cathode surface is several times faster than that on the anode surface.	[86]
Xenopus	neurons nerve duct	DC	1~10 V/cm 7~190 mV/mm	6 h 16–18 h	Neurites facing the cathode grew faster, while those facing the anode grew slower. When the field intensity is 7~190 mV/mm, axons preferentially grow towards the negative or cathode.	[87,88]
Chick	DRG neurons	DC	400~500 mV/mm	3~6 h	Retraction was followed by the re-extension of fibers to a preferred orientation perpendicular to the voltage gradient.	[89]
Rat	hippocampal neurons	DC	28, 80 or 219 mV/mm	24 h	Neurites lay perpendicular to the field after exposure to 28, 80, or 219 mV/mm.	[90]
Rat	cortical neurons	AC	0.5~2000 Hz 80 mV/mm at 0.5 Hz to 2 kHz	24 h 4 Days	The direction of the axon was perpendicular to the adjacent electrode. Promoting axon 3D length growth and alternating electric fields orient large-scale axon alignment in 3D.	[52]
Rat	* FT-NPCs	DC	150 mV/mm	2 h/Day	EFs significantly increase the neuronal differentiation rate of FT-derived NPCs and align neurite outgrowth and promote the length of neurite processes.	[82]
Mouse	PC12	AgNWs/PDMS	240 mV and 20 Hz	120 h	The proliferation rate and axon growth of PC12 cells increased only under electrical stimulation. PC12 cells showed axonal orientation perpendicular to the stretching direction.	[53]
Embryonic rat	hippocampal neurons	DC	0.58~4.73 mV/pm	24 h	The growth cones on the single long process (the putative axon) of cultured hippocampal neurons failed to orient themselves with the electric field applied by focusing, while the growth cones on the short and straight process (the putative dendrite) were oriented toward the cathode.	[91]
Rat	Neural Stem/Progenitor Cells	DC, AC	DC EF:437mV/mm AC EF:46 mV/mm	—	The overall arrangement of NPCs grew obviously perpendicular to the electric field axis.	[92]
Rat	DRG—astrocyte culture	DC	10 mVmm^−1^ or 500 mV mm^−1^	24 h	The growth of neurites was in the same direction and along the same process of the aligned astrocytes.	[93]
Mouse	NPCs/NSCs	DC	150 mV mm^−1^	1 h per day	Induced neuronal differentiation and neurite extension	[94]
Mouse	PC12	DC	0.07 mA	2 h	Electrical stimulation results in longer neurites, more growth, and alignment of cells and neurites at an angle to the applied current.	[95]
Guinea-pig	spinal cord	DC	~100 μV/mm	3 weeks	EF initiate regeneration of central axon in adult guinea pig spinal cord transects.	[96]
Mouse	Primary pre-frontal cortical	two-electrode	±0.25 mA/cm^2^	8 h per 24-h period for 3 days	Electrical stimulation using conductive polymer polypyrrole counters reduced the neurite outgrowth of primary prefrontal cortical neurons from NRG1-KO and DISC1-LI mice.	[97]
Mouse	Primary Cortical Neurons	—	1 ± 0.25 mA/cm^2^	8 h per 24 h period for 3 days	3D electrical stimulation improved the neurite outgrowth in 3D neuronal cultures from both wild-type and NRG1-knockout (NRG1-KO) mice.	[98]

* FT: The adult filum terminale (FT) is an atypical region from where multipotent neural progenitor cells (NPCs) have been isolated.

#### 4.4.1. Electrical Stimulation Promotes Neurites Growing toward the Negative Pole

Electrical stimulation promoting neurite growth was first observed in chicken embryo dorsal root ganglia, and then further identified in neural cells isolated from *Xenopus* or chicken embryonic neural tube. By applying an EF of 70~140 mV/mm to the culture of a chicken embryo dorsal root ganglion, the growth rate of the protrusions facing the cathode was significantly faster compared to those facing the anode [99]. Similar results were obtained from a neuronal *Xenopus* culture: the growth of neurites facing toward the cathode was accelerated, while the growth of neurites facing the anode slowed down [87]. A similar phenotype was also observed in cultured cells from earlier developmental stages. McCaig et al., found that both the neural cells isolated from the embryonic neural tube and *Xenopus* spinal neurites also preferentially grow toward the negative electrode or the cathode, within a range of 7~190 mV/mm of EFs [88]. These results suggest that EFs can be a similar guidance cue of chemotaxis in stimulating neurite outgrowth and orientating them in a specific direction.

Subsequently, various cell types exhibit a parallel arrangement under electric field stimulation in vitro [100].

#### 4.4.2. Axons Exhibit a Parallel Arrangement

Under EF stimulation, in addition to directing neurites growing toward the negative pole, some studies have found that neural processes from both CNS and PNS neurons grow perpendicular to the EF axis within the physiological EF [51,53].

It has been reported that sympathetic neurons from either chicken embryo or rat DRG neurons showed neurites grown perpendicularly to the EF axis [89]. In CNS, the hippocampal, cortical, and spinal cord neurons exhibit extending growth and are oriented in both DC and AC EFs [52,91]. This phenomenon was further confirmed on NSC-derived neurons [51,101] and neural cell lines (e.g., PC12 cells) [53]. When applying an EF to rat embryonic hippocampal neurons, the presumed dendritic or axonal growth cones responded differently: growth cones with short and straight protrusions (presumed to be dendrites) turned to the cathode, while growth cones extending from a single long process (presumed to be axons) of hippocampal neurons failed to be re-oriented to the recently applied electrical field [91]. This result suggested that only axons have the potential to grow perpendicular to the EF axis in the physiological EF. We should also consider that the different responses of neurites in EF are also closely related to intensity, strength, and developmental time points when applying EFs. Higher EF intensity may be required for the vertically parallel growth of electrical stimulation-initiated neurite arrangement, while lower EF intensity may mean that the process takes a much longer time. For instance, an EF of 400–500 mV/mm will take 2–6 h to arrange the neurites to grow perpendicularly to the EF-axis, while with a lower-voltage EF, the time of orientation may take more than 12 h [91].

Besides neurons, other neural cell types, such as NPCs/NSCs and astrocytes, can also be oriented with physiological EF [92,93]. Additionally, the growth of neurites was in the same direction along the aligned astrocytes [93].

#### 4.4.3. Electrical Stimulation Activates Perpendicular Neurite Arrangement in 3D Conditions

Electrical stimulation could also activate perpendicular neurite arrangement in 3D conditions. Therefore, engineered 3D tissue models have been used to test out the effects of different electrical stimulation patterns on axon growth [94,95]; meanwhile, these results further confirmed that it is possible to align the orientation of axons perpendicularly to the EF axis. EF can also initiate the regeneration of the transected central axon in the adult guinea pig spinal cord [96].

Conversely, some types of electrical stimulations may also restore abnormal gene expression, leading to deficits in neurite outgrowth and synaptogenesis. Therefore, optimizing the stimulation protocol would be important before applying EFs to cultured cells. Zhang et al., indicated that electrical stimulation improved neurite outgrowth and the expression of synaptogenesis markers in 3D neuronal cultures derived from both wild-type and NRG1-knockout (NRG1-KO) mice [97,98]. By using a silk protein material-based 3D brain tissue model, an electrode in an AC field could act as an alternating cathode that attracts the growing tip of the axon. In this research, dissociated rat cortical neurons were exposed to an alternating field of 80 mV/mm at 0.5 Hz to 2 kHz, and the 0.5–20 Hz field was found to promote axon growth in a perpendicular orientation in parallel to the field direction [50].

## 5. Electrical Stimulation Potentially Promotes Neuronal Network Formation and Constructing Layer-Specific Connections

As previously mentioned, it could be difficult for neurons in the brain organoids to develop specific hierarchical dendritic structures and axonal projection patterns; thus, it could also be difficult to form myelin sheaths and establish well-developed neural networks [45].

A well-developed neural network is a highly ordered structure with functionally arranged axons and branches of dendrites connected by a massive number of synapses. Its formation includes two sets of events: dendritogenesis and synaptogenesis. Dendritogenesis is concurrent with synaptogenesis [59]. In the developing cortex, it generates ion flow in the vicinity of the electrically mature zone and voltage difference among these different zones. The endogenous electric current could orient cell polarization and eventually drive corresponding morphological changes: axon initiation under the SP and dendritogenesis in the MZ [60]. Recent studies have further identified that the growing thalamic axons need electrical activity to reach the corresponding cortical target area [102,103], while the axons of the cortical pyramidal neurons need electrical activity to form layer-specific connections [103]. Extracellular electrical stimulations could also enhance the synaptic potential of the cortical network coupled to the multi-electrode array [104]. Therefore, modulating synaptic activity and regulating synaptogenesis could help to create tripartite synapses, and adding electrical stimulation could play an important role in neuronal network formation [105,106].

Why are EFs so important for neural circuit formation? One possible reason could be that EF stimulation induces the expression of adhesive molecules, thus increasing cell-cell contact [107]. Moreover, EFs can guide neural processes that extend parallel to the EF axis and promote the branching of neural processes [88]. Ma et al. [69] measured the neurite growth of NSC-derived neurons after ELF-EMF exposure and found that the total length of the neurites and the number of branch points per cell were both significantly increased.

Based on this finding, Meng et al. [94] applied EFs to induce the neuronal differentiation and growth of mice NSCs in 3D Matrigel. With EF stimulation, the 3D-engineered neural tissues produced a large number of synapses, together with the growth of highly branching and orderly neurites. Meanwhile, the formation of the myelin sheath was also observed in the constructed 3D-engineered neural tissues, which is another important structure in the functional neural network of brain organoids (Figure 3).

Electrical stimulation has multiple potential applications in brain organoids by regulating synaptogenesis and dendritogenesis, by stimulating the neurite outgrowth, by increasing the number of branch points, and by orienting the arrangement of axons. However, little research has been conducted into the effect of EFs on cortical layer-specific connections in brain organoids. This may be caused by the limited and small size of cultured brain organoids. As a result, the architecture of the brain organoid is quite different from the normal brain. The formation of microvascular networks has become one of the major bottlenecks that has limited the construction of brain organoids.

## 6. Signaling Pathways Potentially Induced by EFs in Neuronal Cultures

Cell behavior under electrical stimulation has been studied in a variety of cell types, including neural cells [18,50,51,53]. Therefore, their regulatory mechanisms have been extensively explored. Here, we briefly summarize the molecular signaling mechanisms of EF-regulated neural lineage cells.

The activation of the signal transduction pathways is considered to be a possible mechanism for electrical stimulation-regulated cell functions [108]. How does a bioelectrical signal translate into an intracellular signaling system that initiates a series of cellular responses? The widely accepted view is that bioelectrical signals can activate ion channels, some voltage-sensitive genes, or receptors on cell membranes to initiate downstream signaling pathways, thus producing various biological responses (Figure 4) [109,110]. Furthermore, EF-induced cellular responses, such as migration and proliferation, can be coordinated and regulated by multiple signaling pathways [51,63,84]. This raised the possibility that an EF-initiated signaling pathway may have crosstalk with certain chemical-induced signaling cascades.

For instance, both the n-methyl-D-aspartate receptor (NMDAR) and AChR receptors have been shown to be activated at the cathode, inducing Ca^2+^ influx and cell depolarization. EF can affect the NMDAR ligand gate and then activate the NMDAR/Rac1/actin signaling pathway, to regulate cell migration. AChR can induce PI3K to bind to PLC through EF stimulation and activate the CAMP/PKA/Rho pathway. As a result, Rho/Cdc42 regulates cell migration, while the activation of Rho/ROCK promotes actin aggregation and redistribution to growth cones and protuberances. Another study showed that the ion channels, NaKA and NHE3, are the transfer points of cathode/anode migration that induce the inflow of Na^+^/Ca^2+^, which changes the original ion gradients and may adjust the distribution of EFs. This will eventually lead to cell polarization and cytoskeleton redistribution [111,112]. Previous studies have also suggested that EFs may induce EGFR activation in a ligand-independent manner and trigger the downstream MAPKs system, leading to MAPK/ERK activation, cytoskeleton reorganization, and directed cell migration [113,114,115,116].

Zhao et al., reported that using EFs in the process of wound repair involves activating a voltage-sensitive gene, Phosphatidylinositol 3-kinase (PI3K), to make cells respond to electrical stimulation with electrotaxis [117]. Subsequent studies have further demonstrated that during NSCs proliferation, EFs and fibroblast growth factor (bFGF) synergistically mediated the directional migration of NSCs through the activation of the PI3K/AKT signaling pathway [63]. When the NSCs halt cell proliferation and initiate cell differentiation, PI3K/AKT signaling crosstalks with the Wnt/GSK-3β/β-catenin signaling pathway through GSK-3β, promoting the accumulation of β-catenin and the continuous expression of Ascl1, ultimately determining the differentiation of cells into neurons [51]. In another study, the expression of the early neuronal gene ENO2 and MECP2 was increased in NSCs when they were stimulated by EFs (5 mV, 0.5 mA, 25 ms intermittent stimulation) [118].

## 7. Conclusions and Future Prospects

This review summarizes the types of brain organoids and the methods currently used in the lab and discusses the potential role of EF in the construction of brain organoids. The in vitro construction of brain organoids is a process that represents in vivo neural development. The pivotal events in reproducing brain organoids may include: (1) direct neural differentiation in organoids; (2) when neurons in the organoids form highly branched and ordered neurites; (3) the formation of a complete myelin sheath; (4) the formation of a well-developed neuronal network.

As an in vitro tissue model simulating brain structures, brain organoids are still far from able to replicate all brain developmental features and organizational structures. Currently, the drawbacks of using brain organoids include: (1) lacking sufficient and precise functional neuron subtypes; (2) the neurons in organoids lack highly branched and ordered neural processes; (3) lacking mature oligodendrocytes, resulting in the absence of a complete myelin sheath; (4) the difficulty of mimicking brain development in the late embryo and postnatal periods; (5) the reproducibility, accuracy, regionalization, and vascularization of brain organs.

Electrical stimulation exhibits interesting potential in regulating or even controlling the following events: (1) EF can induce the neuronal differentiation of NSCs, which is a forerunner of the formation of brain organoids; (2) EF can promote the parallel and ordered growth of neural processes in 3D-cultured neural tissues; (3) EF can promote the branching of neural processes; (4) EF can promote the formation of synapses and myelin sheaths in a 3D environment, and then create a well-developed 3D neural network. Therefore, the synergistic effect of EF and other factors may play a critical role in the construction of brain organoids. However, in the process of continuous practice, brain organoid construction still needs further research to investigate EF intensity, the time point, and the operational procedure of EF application in 3D cultures; its synergistic effect with biochemical factors and the conductivity of biomaterials could be a promising future research area.

## Figures and Tables

**Figure 1 ijms-23-03877-f001:**
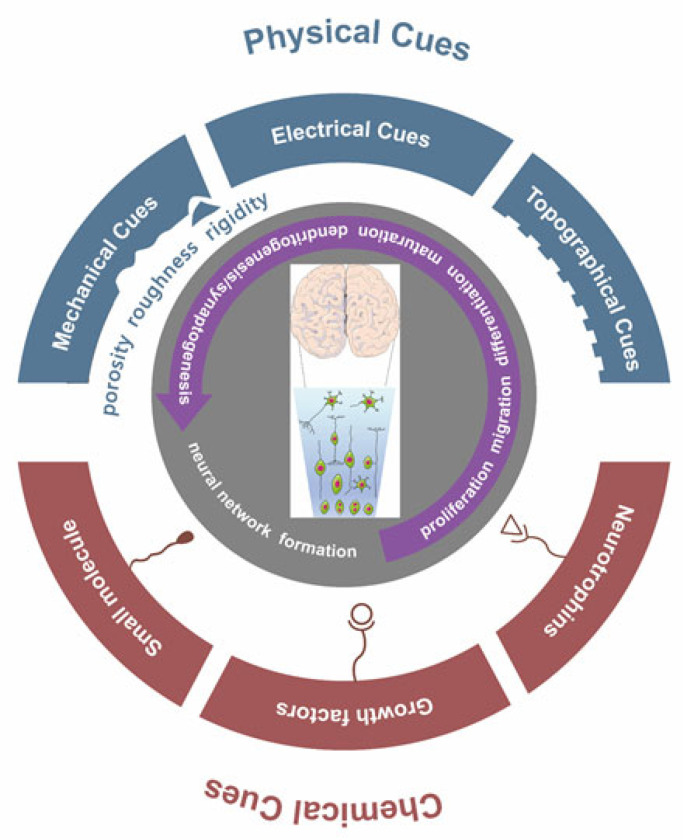
Chemical signals and physical cues work together to regulate the developmental process of neural tissue. In nervous system development, intrinsic (e.g., transcription factors) and extrinsic (e.g., environmental signals) cues cooperate to regulate neuronal network formation and tissue construction. Both chemical and physical signals in the microenvironment influence cell behaviors and tissue formation. Chemical cues, such as growth factors, hormones, neurotrophins (NTFs), and the extracellular matrix (ECM) influence neuronal fate decisions, cell migration and neurite maturation. Physical cues, such as mechanical cues (e.g., rigidity), electrical, and topographical cues in extracellular environments synergistically interact with these molecular cues.

**Figure 2 ijms-23-03877-f002:**
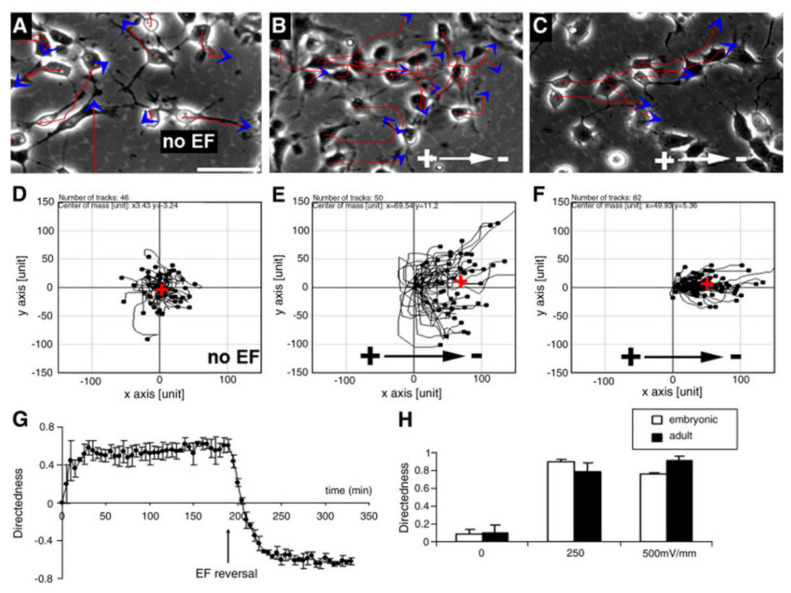
eNPCs and aNPCs show directed migration in EFs. (**A**–**C**) Migration of eNPCs (**A**,**B**) and aNPCs (**C**) in the absence (**A**) or presence of EFs (**B**,**C**). Red lines and blue arrows represent the trajectory and direction of cell movement. (**D**–**F**) Migration paths of a group of cells plotted as if all starting from the origin position, either without EFs (**D**), or in 500 mV/mm ((**E**,**F**) show the migration paths of eNPCs and aNPCs separately). The red cross in (**D**–**F**) represents the center of mass of all cell ending positions, which indicates to what extent directed cell migration occurred, in terms of direction and efficiency. (**G**) shows a sharp reversal in the direction of migration after EF polarity. (**H**) Directedness as a function of EF strength. Scale bar: 50 µm (cited from our previous work, reprinted unaltered from reference [64] (https://doi.org/10.1016/j.expneurol.2010.11.002).

**Figure 3 ijms-23-03877-f003:**
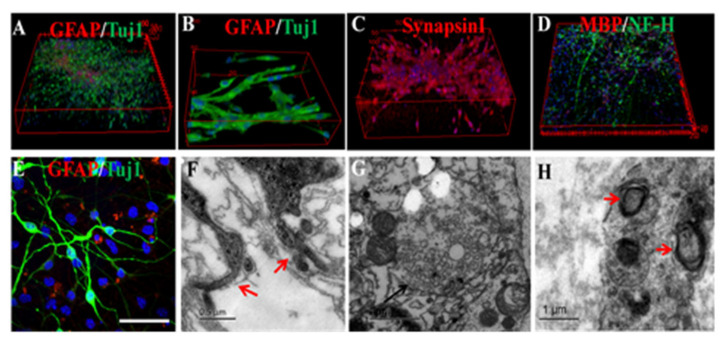
EFs induced NSC neuronal differentiation and neurite extension in 3D-constructed Matrigel. Representative images of 3D scans (**A**) and 2D scans (**E**) of neural constructs. Electrical stimulation produced highly branched neurites and a well-developed neuronal network (**B**,**F**). SynapsinI immunoreactivity was observed around the soma and neurites (**C**), and MBP was detected in the culture (**D**). Results of ultrastructural imaging of the formative synapse (**G**) and myelin sheath ((**H**), red arrow) in engineered neural tissue. Scale bar: E, 50 µm; the scale bars of the ultrastructural image represent 0.5 or 1 µm. (Cited from our previous work, reproduced from (Reference [95], https://iopscience.iop.org/article/10.1088/1741-2552/abaac0, accessed on 28 October 2020)).

**Figure 4 ijms-23-03877-f004:**
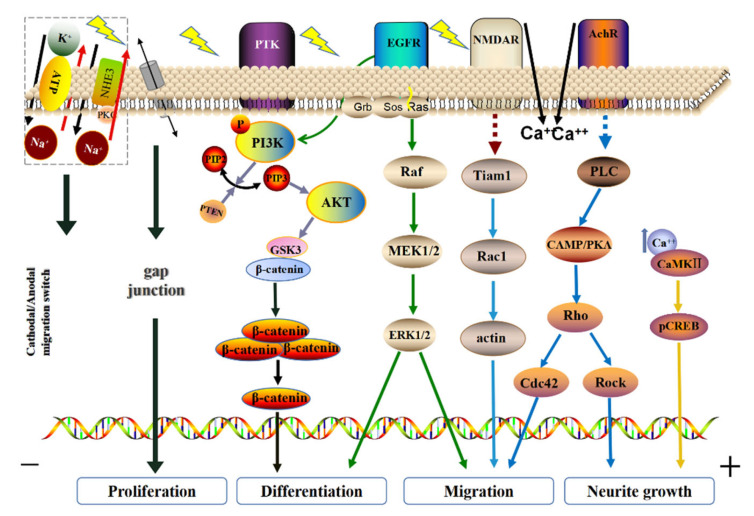
Bioelectrical signals can activate ion channels, some voltage-sensitive genes, or receptors on cell membranes to initiate the downstream signaling pathways, thus producing various biological responses. The influx of Na^+^ and Ca^2+^ leads to the formation of ion gradients along or against the direction of EF. This will eventually lead to cell polarization and cytoskeleton redistribution. EF stimulation also induces EGFR activation in a ligand-independent manner. This triggers the downstream MAPK system, which leads to MAPK/ERK activation, cytoskeleton reorganization, and directed migration. When NSCs halt cell proliferation and initiate cell differentiation, PI3K/AKT signaling has crosstalk with the Wnt/GSK-3β/β-catenin signaling pathway through GSK-3β, promoting the accumulation of β-catenin and ultimately determining the differentiation of cells into neurons. EF can affect the NMDAR ligand gate and then activate the NMDAR/Rac1/actin signaling pathway to regulate cell migration. EF-stimulated AChR regulates directional cell migration and promotes neurite extension through different downstream signaling pathways.

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
