# Peer review of "Physiological Electric Field: A Potential Construction Regulator of Human Brain Organoids"

_ijms, 2022, doi:10.3390/ijms23073877_

Round 1

Reviewer 1 Report

In this review, authors present a comprehensive study by focusing on the effects of physiological electric field on the regulation of brainoid formation. Reviewer thinks that this is a novel and useful review for researchers in the field. It can be published after the execution of the following revisions:

General minor 1: Put comma before “and” if there are more than 2 items as a list.

General minor 2: Check grammar, especially the usage of past tense. Check capital letter usage.

Section 2.

  • To obtain more systematic and fluent organization, section 2 should be presented as 2 different main section: section 2 and new section 3. Section 2 can have a title like “traditional approaches to obtain brainoid” and new section 3 can have the same title with old section 2 “Whether adding EFs could enhance the efficacy of constructing Human Brain Organoids”. Moreover, subsection 2.1 and 2.2 should stay under section 2 while section 2.3 should be the new section 3.1.
  • Additionally, traditional brainoid construction approaches section reference number should be increased by newer references from 2020 and 2021.
  • Change the following sentence: “This section may be divided by subheadings” as “This section is divided by subheadings.”
  • The following sentence requires a reference and please check whether 5cm-1cm is accurate. “The size of cultured brain organoids is also tiny: their dimensions are generally only around 5cm-1cm.”
  • Correct the following sentence: “Neurons in organoids lack specific hierarchical..” as “…lack of..”.
  • Figure 1: First, the following sentences in the title should be taken away. “This is a figure. Schemes follow the same formatting”. The figure itself is too basic for this type of comprehensive review. Make it more presentable and more informative. You may add some examples of cues under the main titles on the figure (not only in the title).

Section 3.

  • Old section 3 should be the subsection of new section 3 to provide fluent reading.
  • Correct the sentence “Endogenous EFs not only..” as “Endogenous Efs are not only..”
  • In Table 1, if there is 2 different main results from one reference, merge them to one reference line. Presenting them in 2 different lines makes Table 1 more complicated.

Section 4.

  • This section can stay as a section 4 (major section).
  • Please check the following sentence: “Endogenous EFs generated ions flows”. Is it endogenous or exogenous?
  • If possible, it could be more informative if some visuals are inserted into section 4 as 1 figure. For example, they can represent some electrotaxis images.
  • Please re-write the following sentence which is also not explicit in the main reference: “The proliferation rates and axon growths of PC12 cells can be enhanced by electrical stimulations, while slightly enhancing the intensity of electrical stimulations increases the number of cells significantly”.
  • In table 2, parameters column has different type of parameters. Could you please make it more comprehensive like table 3? For example, for some signals, even it has frequency value, it does not have voltage/ current magnitude values. Presenting electric field strength could be more useful and systematic.

Section 5.

  • Ref 104 has explanatory images. If copyright issues are not problematic for you, including those images will increase the value of this review paper.
  • There are not so many studies to interpret/discuss effects of EFs on layer formation in 3D cell culture and brain organoids according to section 5. Could you please elaborate on “why”? What are the challenges behind this?

Section 6.

  • This sentence requires some new references: “Cell behaviours under electrical stimulations has been studied in a variety of cell types including neural cells.”

Reviewer 2 Report

Current review “Physiological Electric Field: A potential construction regulator of human brain organoids” by Yu et al. excellently described about human brain organoids and the importance of physiological electric field. Authors elaborated on the use of physiological electric field in successful development of brain organoid as it is one of the important signaling cue in in vivo brain functions.

Comments:

  1. In the introduction section Authors need to elaborate briefly (single paragraph) on the historical development of brain organoids and how the methods were evolved/evolving by improving the assay conditions such as media composition which is part of chemical cues and then physical cues and others (mechanical/strain/stiffness by extracellular matrix components etc.).
  2. Line 184 Please remove “this is a figure”
